# Finite Pure Plane Strain Bending of Inhomogeneous Anisotropic Sheets

**Sergei Alexandrov** [1] **, Elena Lyamina** [1] **and Yeong-Maw Hwang** [2,*]

1 Laboratory for Technological Processes, Ishlinsky Institute for Problems in Mechanics RAS, 119526 Moscow, Russia; sergei_alexandrov@spartak.ru (S.A.); lyamina@inbox.ru (E.L.)
2 Department of Mechanical and Electro-Mechanical Engineering, National Sun Yat-Sen University, Kaohsiung 80424, Taiwan
* Correspondence: ymhwang@mail.nsysu.edu.tw

**Abstract:** The present paper concerns the general solution for finite plane strain pure bending of incompressible, orthotropic sheets. In contrast to available solutions, the new solution is valid for inhomogeneous distributions of plastic properties. The solution is semi-analytic. A numerical treatment is only necessary for solving transcendent equations and evaluating ordinary integrals. The solution's starting point is a transformation between Eulerian and Lagrangian coordinates that is valid for a wide class of constitutive equations. The symmetric distribution relative to the center line of the sheet is separately treated where it is advantageous. It is shown that this type of symmetry simplifies the solution. Hill's quadratic yield criterion is adopted. Both elastic/plastic and rigid/plastic solutions are derived. Elastic unloading is also considered, and it is shown that reverse plastic yielding occurs at a relatively large inside radius. An illustrative example uses real experimental data. The distribution of plastic properties is symmetric in this example. It is shown that the difference between the elastic/plastic and rigid/plastic solutions is negligible, except at the very beginning of the process. However, the rigid/plastic solution is much simpler and, therefore, can be recommended for practical use at large strains, including calculating the residual stresses.

**Keywords:** plastic anisotropy; large strain; pure bending; elastic unloading

## 1. Introduction

Sheet metal forming processes usually include bending. A brief review of typical sheet metal forming processes that incorporate bending is provided in [1]. The process of bending is also an essential test for identifying material properties, for example [2–10]. Theoretical analyses of the bending process are necessary for interpreting test results.

An exact rigid perfectly plastic solution for finite pure plane strain bending and plane strain bending under tension of sheets has been found in [11]. This solution has been adopted in [12] for deriving a closed form expression for strain at any fiber. This paper has concluded that the basic assumptions made in [11] are plausible. The solution [11] has been extended to many material models. Of primary interest for the present paper are solutions for anisotropic materials and inhomogeneous sheets. A brief review of such solutions is given below.

Elastic and plastic anisotropy is a typical property of many metallic and non-metallic materials [13–16]. A solution for pure plane strain bending of anisotropic sheets has been derived in [17]. The model adopted incorporates the Bauschinger effect and strain hardening. Plastic anisotropy is described by Hill's quadratic yield criterion [11]. It has been found that the effect of plastic anisotropy on the bending moment is rather significant. Papers [18,19] present an analysis of elastic/plastic bending of orthotropic sheets. Elastic properties are isotropic, and plastic yielding obeys Hill's quadratic yield criterion [11]. A more detailed review of solutions for plane strain bending of anisotropic sheets is provided in [20]. An extension of the solutions above has been proposed in [21] where

tension/compression asymmetry of plastic properties has been considered. All these solutions are for homogeneous sheets.

Several solutions are available for piecewise homogenous sheets. The pure bending of bonded laminated metals under plane strain conditions has been considered in [22,23]. Different rigid plastic material models have been adopted in these papers. Elastic properties have been taken into consideration in [24]. This paper emphasizes the prediction of springback. The distribution of residual stresses in bilayer sheets after bending has been found in [25]. A simplified solution for bending and subsequent unloading of bilayer sheets has been proposed in [26].

Different methods have been employed to get the solutions above. A unified approach for analyzing finite pure plane strain bending has been developed in [27]. The approach applies to a broad class of incompressible materials. In particular, the corresponding solutions for anisotropic and bilayer sheets have been found in [28] and [29], respectively.

In many cases, the through-thickness distribution of material properties in sheets is non-uniform but, in contrast to [22–26], is described by continuous functions [30]. Using the approach [27], the pure bending of isotropic elastic/plastic functionally graded sheets has been analyzed in [31]. The present paper extends this solution to anisotropic sheets. In addition, a rigid/plastic model is considered. The general solution involves quite an arbitrary through-thickness distribution of plastic properties. The numerical example is based on the experimental data presented in [32]. In the case of the rigid/plastic solution, the prediction of residual stresses can be made using the methodology proposed in [33] and further developed and discussed in [34]. According to this methodology, for computing residual stresses in a deformation process, the elasticity is neglected during the process's loading phase. It is shown that the solution for purely elastic unloading is not valid even for a relatively large inside radius.

## 2. Statement of the Problem

A metallic sheet is bent under plane strain conditions by two couples *M*, as shown in Figure 1. The initial shape of the sheet in the planes of flow is a rectangular of thickness *H* and width 2*L*.

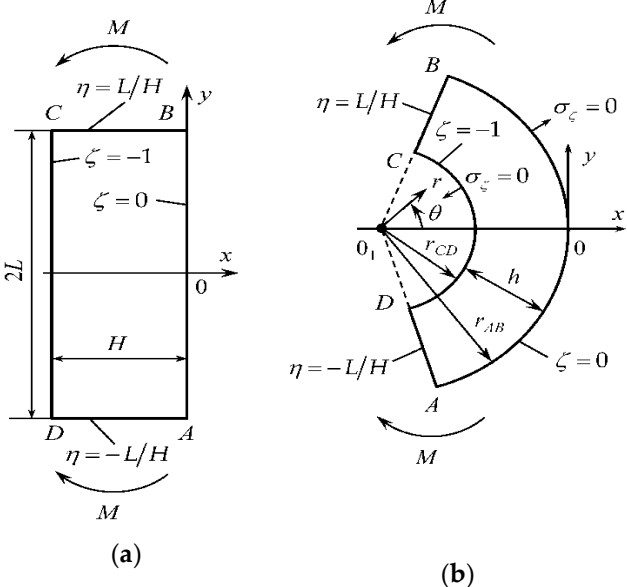

**Figure 1.** Schematic diagram of the pure bending process: (**a**) initial configuration, (**b**) intermediate and final configurations.

Curves *AB* and *CD* are circular arcs, and *AD* and *BC* are straight throughout the process of deformation (except the initial instant when *AB* and *CD* are straight). It has been

shown in [27] that the following equations describe the transformation of the initial shape into the shape after any amount of strain:

$$\frac{x}{H} = \sqrt{\frac{\zeta}{a} + \frac{s}{a^2}} \cos(2a\eta) - \frac{\sqrt{s}}{a} \quad \text{and} \quad \frac{y}{H} = \sqrt{\frac{\zeta}{a} + \frac{s}{a^2}} \sin(2a\eta). \tag{1}$$

Here $(x, y)$ are Eulerian Cartesian coordinates and $(\zeta, \eta)$ are Lagrangian coordinates. Additionally, $a$ is a time-like variable such that $a = 0$ at the initial instant and $s$ is a function of $a$. The latter should be found from the solution. The $x$-axis is an axis of symmetry of the process. The Cartesian coordinate system's origin is situated at the intersection of the axis of symmetry and curve $AB$. At the initial instant,

$$x = \zeta H \quad \text{and} \quad y = \eta H. \tag{2}$$

This condition is satisfied if

$$s = \frac{1}{4} \tag{3}$$

at $a = 0$. It is convenient to introduce a plane polar coordinate system $(r, \theta)$ with the origin at $x = -H\sqrt{s}/a$ and $y = 0$. It follows from (1) that

$$\frac{r}{H} = \frac{\sqrt{\zeta a + s}}{a} \quad \text{and} \quad \theta = 2a\eta. \tag{4}$$

It is seen from (2) and Figure 1 that $\zeta = 0$ on curve $AB$ and $\zeta = -1$ on curve $CD$ throughout the process of deformation. The radii of circular arcs $AB$ and $CD$ are determined from (4) as

$$\frac{r_{AB}}{H} = \frac{\sqrt{s}}{a} \quad \text{and} \quad \frac{r_{CD}}{H} = \frac{\sqrt{s - a}}{a}, \tag{5}$$

respectively. The transformation (1) satisfies the equation of incompressibility.

Let $\sigma_\zeta$ and $\sigma_\eta$ be the normal stresses referred to the Lagrangian coordinate system. This coordinate system is orthogonal, and the normal stresses $\sigma_\zeta$ and $\sigma_\eta$ are the principal stresses. The principal axes of anisotropy coincide with the $x$—and $y$—axes at the initial instant. Then, according to the model proposed in [35], the principal anisotropy axes coincide with the $\zeta-$ and $\eta-$ coordinate curves throughout the process of deformation. Under plane strain conditions, the anisotropic yield criterion proposed in [11] reads

$$\left| \sigma_\eta - \sigma_\zeta \right| = 2T\sqrt{1 - c} \tag{6}$$

Here $T$ is the shear yield stress with respect to the $\zeta-$ and $\eta-$ coordinate curves, and $c$ is expressed through the yield stresses in the principal anisotropy axes' directions [11]. Both $T$ and $c$ depend on $\zeta$. It is convenient to represent $T$ as $T = T_0\omega(\zeta)$ where $T_0$ is constant with the dimensions of stress. Then, Equation (6) becomes

$$\left| \sigma_\eta - \sigma_\zeta \right| = T_0 W(\zeta) \tag{7}$$

where

$$W(\zeta) = 2\omega(\zeta)\sqrt{1 - c}. \tag{8}$$

The transformation equations between the $(x, y)-$ and $(\zeta, \eta)-$ coordinate systems in (1) satisfy the flow rule associated with the yield criterion (7).

Let $\xi_\zeta$ and $\xi_\eta$ be the principal components of the total strain rate tensor. This tensor is the sum of the elastic and plastic strain rate tensors. Then,

$$\xi_\zeta = \xi_\zeta^e + \xi_\zeta^p \quad \text{and} \quad \xi_\eta = \xi_\eta^e + \xi_\eta^p. \tag{9}$$

The superscript *e* denotes the elastic portion of the strain rate components and the superscript *p* the plastic portion. The elastic portion is related to the stress components as

$$\dot{\sigma}_\zeta - \dot{\sigma} = 2G\xi_\zeta^{e} \quad \text{and} \quad \dot{\sigma}_\eta - \dot{\sigma} = 2G\xi_\eta^{e}. \tag{10}$$

Here the superimposed dot denotes the convected derivative, $\sigma$ is the hydrostatic stress and $G$ is the shear modulus of elasticity. Since the material is incompressible, $\sigma = (\sigma_\zeta + \sigma_\eta)/2$. Then, Equation (10) becomes

$$\dot{\sigma}_\zeta - \dot{\sigma}_\eta = 4G\xi_\zeta^{e} \quad \text{and} \quad \dot{\sigma}_\eta - \dot{\sigma}_\zeta = 4G\xi_\eta^{e} \tag{11}$$

The only stress equilibrium equation which is not identically satisfied in the polar coordinate system is

$$\frac{\partial \sigma_r}{\partial r} + \frac{\sigma_r - \sigma_\theta}{r} = 0. \tag{12}$$

It is evident from (4) that $\sigma_r = \sigma_\zeta$ and $\sigma_\theta = \sigma_\eta$. Therefore, Equation (12) becomes
The boundary conditions are

$$\frac{\partial \sigma_\zeta}{\partial r} + \frac{\sigma_\zeta - \sigma_\eta}{r} = 0. \tag{13}$$

$$\sigma_\zeta = 0 \tag{14}$$

for $r = r_{AB}$ (or $\zeta = 0$) and $r = r_{CD}$ (or $\zeta = -1$).
The bending moment is determined as

$$M = \int_{r_{CD}}^{r_{AB}} \sigma_\eta r dr. \tag{15}$$

## 3. General Solution

Using (1), one can immediately find the total principal strains as

$$\varepsilon_\zeta = -\frac{1}{2} \ln[4(\zeta a + s)] \quad \text{and} \quad \varepsilon_\eta = \frac{1}{2} \ln[4(\zeta a + s)]. \tag{16}$$

Using (4), one transforms Equation (13) to

$$\frac{\partial \sigma_\zeta}{\partial \zeta} + \frac{a(\sigma_\zeta - \sigma_\eta)}{2(\zeta a + s)} = 0 \tag{17}$$

and Equation (15) to

$$m = \frac{4M}{T_0 H^2} = \frac{2}{a} \int_{-1}^{0} \frac{\sigma_\eta}{T_0} d\zeta. \tag{18}$$

It is worthy of note that $m = 1$ if $W(\zeta) = 1$ in (7), as follows from [11].

### 3.1. Purely Elastic Solution

In the case of the purely elastic solution, $\xi_\zeta^{e} = \xi_\zeta$ and $\xi_\eta^{e} = \xi_\eta$ in (11). The solution of Equations (11) and (17) supplemented with the equation of strain compatibility is [36]

$$\frac{\sigma_\zeta}{T_0} = \frac{1}{2k} \ln^2[4(\zeta a + s)] + C, \quad \frac{\sigma_\eta}{T_0} = \frac{1}{2k} \ln^2[4(\zeta a + s)] + \frac{2}{k} \ln[4(\zeta a + s)] + C. \tag{19}$$

Here $k = T_0/G$ and $C$ is constant.

*3.2. Solution in Plastic Regions Where $\sigma_\eta > \sigma_\zeta$*

In this case, the yield criterion (7) becomes

$$\sigma_\eta - \sigma_\zeta = T_0 W(\zeta). \tag{20}$$

Substituting this equation into (17) leads to

$$\frac{\partial \sigma_\zeta}{T_0 \partial \zeta} - \frac{aW(\zeta)}{2(\zeta a + s)} = 0. \tag{21}$$

One integrates this equation to get

$$\frac{\sigma_\zeta}{T_0} = \frac{a}{2} \int_{\zeta^{(1)}}^{\zeta} \frac{W(\mu)}{(\mu a + s)} d\mu + \frac{\sigma_\zeta^{(1)}}{T_0}. \tag{22}$$

Here $\mu$ is a dummy variable of integration and

$$\sigma_\zeta = \sigma_\zeta^{(1)}. \tag{23}$$

at $\zeta = \zeta^{(1)}$. Equations (20) and (22) combine to give

$$\frac{\sigma_\eta}{T_0} = \frac{a}{2} \int_{\zeta^{(1)}}^{\zeta} \frac{W(\mu)}{(\mu a + s)} d\mu + W(\zeta) + \frac{\sigma_\zeta^{(1)}}{T_0}. \tag{24}$$

*3.3. Solution in Plastic Regions Where $\sigma_\eta < \sigma_\zeta$*

In this case, the yield criterion (7) becomes

$$\sigma_\zeta - \sigma_\eta = T_0 W(\zeta). \tag{25}$$

Substituting this equation into (17) leads to

$$\frac{\partial \sigma_\zeta}{T_0 \partial \zeta} + \frac{aW(\zeta)}{2(\zeta a + s)} = 0. \tag{26}$$

One integrates this equation to get

$$\frac{\sigma_\zeta}{T_0} = \frac{a}{2} \int_{\zeta}^{\zeta^{(2)}} \frac{W(\mu)}{(\mu a + s)} d\mu + \frac{\sigma_\zeta^{(2)}}{T_0}, \tag{27}$$

where

$$\sigma_\zeta = \sigma_\zeta^{(2)} \tag{28}$$

at $\zeta = \zeta^{(2)}$. Equations (25) and (27) combine to give

$$\frac{\sigma_\eta}{T_0} = \frac{a}{2} \int_{\zeta}^{\zeta^{(2)}} \frac{W(\mu)}{(\mu a + s)} d\mu - W(\zeta) + \frac{\sigma_\zeta^{(2)}}{T_0}. \tag{29}$$

## 4. Initiation of Plastic Yielding

The entire sheet is elastic at the beginning of the process. In this case, the solution (19) is valid in the range $-1 \leq \zeta \leq 0$. Therefore, this solution should satisfy the boundary conditions in Equation (14). Then,

$$\frac{1}{2k} \ln^2(4s) + C = 0 \quad \text{and} \quad \frac{1}{2k} \ln^2[4(s-a)] + C = 0. \tag{30}$$

Eliminating $C$ between these equations yields

$$16s(s-a) = 1. \tag{31}$$

Solving this equation for $s$ and then using any of the equations in (30), one gets

$$s = \frac{2a + \sqrt{4a^2 + 1}}{4}, \quad C = -\frac{1}{2k} \ln^2\left(2a + \sqrt{4a^2 + 1}\right). \tag{32}$$

Substituting (32) into (19) supplies the principal stresses' through-thickness distribution at any value of $a$. This solution is valid if the yield criterion is not violated in the range $-1 \leq \zeta \leq 0$. Equations (7) and (19) combine to give

$$-\frac{k}{2} \leq \Phi(\zeta) \leq \frac{k}{2} \tag{33}$$

where

$$\Phi(\zeta) = \frac{\ln[4(\zeta a + s)]}{W(\zeta)}. \tag{34}$$

If $\Phi(\zeta)$ is a monotonically increasing or decreasing function of its argument, then the yield criterion may violate at $\zeta = 0$, or $\zeta = -1$, or $\zeta = 0$ and $\zeta = -1$ simultaneously. The corresponding conditions are

$$\frac{\ln(4s)}{W(0)} = \pm\frac{k}{2} \,, \text{ or } \frac{\ln[4(s-a)]}{W(-1)} = \mp\frac{k}{2} \,, \text{ or } \frac{\ln(4s)}{W(0)} = \pm\frac{k}{2} \text{ and } \frac{\ln[4(s-a)]}{W(-1)} = \mp\frac{k}{2}. \tag{35}$$

Here the upper sign corresponds to monotonically increasing functions, and the lower sign to monotonically decreasing functions.

Let $a_e$ and $s_e$ be the values of $a$ and $s$, respectively, that correspond to plastic yielding initiation. These values are readily found from (32) and (35) if the function $W(\zeta)$ is prescribed. In particular, if the distribution of material properties is symmetric relative to the surface $\zeta = -1/2$, then $W(0) = W(-1)$ and it follows from the third case in (35) that $16s_e(s_e - a_e) = 1$. The latter equation coincides with (31). Therefore, if the distribution of material properties is symmetric relative to the surface $\zeta = -1/2$, then the initiation of plastic yielding occurs at $\zeta = 0$ and $\zeta = -1$ simultaneously.

If the $\Phi(\zeta)$ has a local minimum or maximum, then one should consider the possibility of the initiation of plastic yielding at such points, in addition to the points $\zeta = 0$ and $\zeta = -1$. The initiation of plastic yielding at $\zeta = \zeta_m$ where $-1 < \zeta_m < 0$ occurs if

$$\Phi'(\zeta_m) = 0 \text{ and } |\sigma_\eta - \sigma_\zeta| = T_0 W(\zeta_m). \tag{36}$$

Here $\Phi'(\zeta) \equiv d\Phi/d\zeta$. Using (19) and (34), one transforms Equation (36) to

$$aW\left(\zeta_m^{(1)}\right) - \left(\zeta_m^{(1)}a + s\right)W'\left(\zeta_m^{(1)}\right)\ln\left[4\left(\zeta_m^{(1)}a + s\right)\right] = 0 \text{ and } \ln\left[4\left(\zeta_m^{(1)}a + s\right)\right] = \tfrac{k}{2}W\left(\zeta_m^{(1)}\right) \tag{37}$$

or

$$aW\left(\zeta_m^{(2)}\right) - \left(\zeta_m^{(2)}a + s\right)W'\left(\zeta_m^{(2)}\right)\ln\left[4\left(\zeta_m^{(2)}a + s\right)\right] = 0 \text{ and } \ln\left[4\left(\zeta_m^{(2)}a + s\right)\right] = -\tfrac{k}{2}W\left(\zeta_m^{(2)}\right) \tag{38}$$

Here $W'(\zeta) \equiv dW/d\zeta$. The equations in (37) combine to give

$$a_e^{(1)} = \frac{k}{8} W'\left(\zeta_m^{(1)}\right) \exp\left[\frac{k}{2} W\left(\zeta_m^{(1)}\right)\right]. \tag{39}$$

Then, Equation (32) supplies $s = s_e^{(1)}$ as a function of $\zeta_m^{(1)}$. One can replace $a$ and $s$ in the second equation in (37) with $a_e^{(1)}$ and $s_e^{(1)}$, respectively, to arrive at the equation for $\zeta_m^{(1)}$. This equation should be solved numerically. Then, $a_e^{(1)}$ is readily determined from (39).

One applies the same procedure to (38) for determining $a_e^{(2)}$, $s_e^{(2)}$, and $\zeta_e^{(2)}$. In particular,

$$a_e^{(2)} = -\frac{k}{8} W'\left(\zeta_m^{(2)}\right) \exp\left[-\frac{k}{2} W\left(\zeta_m^{(2)}\right)\right]. \tag{40}$$

If $-1 < \zeta_e^{(1)} < 0$ and $-1 < \zeta_e^{(2)} < 0$, then $a_e = \min\left\{a_e^{(1)}, a_e^{(2)}\right\}$.

In what follows, it is assumed that $a \geq a_e$.

## 5. Rigid Plastic Solution

For many applications, it is possible to assume that the inelastic behavior is dominant and to neglect the elastic response [37,38]. In this case, the general solution given in Section 3 simplifies. There are two plastic regions throughout the process of deformation. The inequality $\sigma_\zeta > \sigma_\eta$ is valid in one of these regions, and the inequality $\sigma_\zeta < \sigma_\eta$ in the other. The neutral line separates the plastic regions. The stress $\sigma_\eta$ is discontinuous across the neutral line.

Let $\zeta = \zeta_n$ be the neutral line. It has been shown in [27] that

$$\zeta_n = -\frac{ds}{da}. \tag{41}$$

The solution (27) is valid in the region $-1 \leq \zeta \leq \zeta_n$. This solution should satisfy the boundary condition (14) at $\zeta = -1$. Then, $\zeta^{(2)} = -1$ and $\sigma_\zeta^{(2)} = 0$ in (27), and this equation becomes

$$\frac{\sigma_\zeta}{T_0} = -\frac{a}{2} \int_{-1}^{\zeta} \frac{W(\mu)}{(\mu a + s)} d\mu. \tag{42}$$

Consequently, Equation (29) becomes

$$\frac{\sigma_\eta}{T_0} = -\frac{a}{2} \int_{-1}^{\zeta} \frac{W(\mu)}{(\mu a + s)} d\mu - W(\zeta). \tag{43}$$

The solution (22) is valid in the region $\zeta_n \leq \zeta \leq 0$. This solution should satisfy the boundary condition (14) at $\zeta = 0$. Then, $\zeta^{(1)} = 0$ and $\sigma_\zeta^{(1)} = 0$ in (22), and this equation becomes

$$\frac{\sigma_\zeta}{T_0} = \frac{a}{2} \int_{0}^{\zeta} \frac{W(\mu)}{(\mu a + s)} d\mu. \tag{44}$$

Consequently, Equation (24) becomes

$$\frac{\sigma_\eta}{T_0} = \frac{a}{2} \int_{0}^{\zeta} \frac{W(\mu)}{(\mu a + s)} d\mu + W(\zeta). \tag{45}$$

The stress $\sigma_\zeta$ must be continuous across the neutral line. Then, Equations (41), (42) and (44) combine to give

$$\int_{-ds/da}^{0} \frac{W(\mu)}{(\mu a + s)} d\mu = \int_{-1}^{-ds/da} \frac{W(\mu)}{(\mu a + s)} d\mu. \tag{46}$$

Since $W$ is a known function of its argument, this equation is an ordinary differential equation for determining $s$ as a function of $a$. However, its form is non-standard. To develop a numerical method for solving (46), it is advantageous that one considers the initial instant separately. The distribution of the stress $\sigma_\eta$ at the initial instant is illustrated in Figure 2. The stress $\sigma_\zeta$ vanishes everywhere.

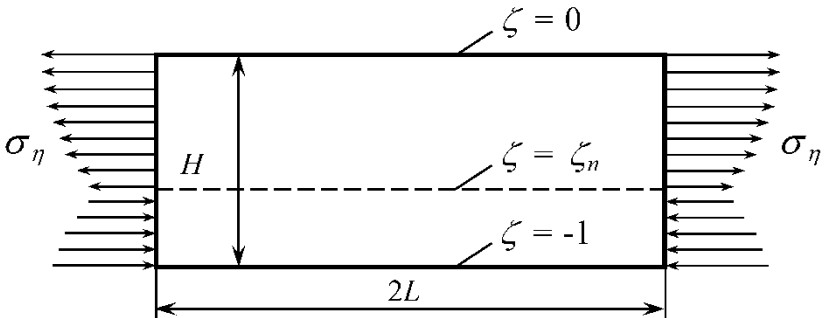

**Figure 2.** Distribution of the stress $\sigma_\eta$ at the initial instant.

Then, it follows from (7) that

$$\sigma_\eta = \begin{cases} T_0 W(\zeta) & \text{in the region } \zeta_n \le \zeta \le 0 \\ -T_0 W(\zeta) & \text{in the region } -1 \le \zeta \le \zeta_n. \end{cases} \tag{47}$$

In the case of pure bending, equilibrium demands

$$\int_{-1}^{0} \sigma_\eta d\zeta = 0. \tag{48}$$

Substituting (47) into (48) and using (41) yields

$$\int_{-ds/da}^{0} W(\zeta) d\zeta - \int_{-1}^{-ds/da} W(\zeta) d\zeta = 0. \tag{49}$$

This equation should be solved for $ds/da$. In general, a numerical method should be used. However, in some cases, an analytic solution is available. For example, if $W$ is constant then $ds/da = 1/2$ at the initial instant. If the distribution of material properties is symmetric relative to the central plane of the sheet, then $W_Z(Z)$ is an even function of $Z$ where $Z = \zeta + 1/2$ and $W_Z(Z) = W(\zeta) = W(Z - 1/2)$. In this case, Equation (49) becomes

$$\int_{-ds/da+1/2}^{1/2} W_Z(Z) dZ - \int_{-1/2}^{-ds/da+1/2} W(\zeta) d\zeta = 0. \tag{50}$$

Integrating gives

$$\Omega(1/2) - 2\Omega(-ds/da + 1/2) + \Omega(-1/2) = 0 \tag{51}$$

where $\Omega(Z)$ is the anti-derivative of $W_Z(Z)$. Therefore, $\Omega(Z)$ is an odd function of $Z$ and $\Omega(1/2) + \Omega(-1/2) = 0$. Then, it follows from (51) that

$$\frac{ds}{da} = \frac{1}{2} \tag{52}$$

at the initial instant.

Using (47), the bending moment at the initial instant is determined as

$$M = T_0 \int_{-ds/da}^{0} W(\zeta)\left(\zeta + \frac{ds}{da}\right) d\zeta + T_0 \int_{-1}^{-ds/da} W(\zeta)\left(\zeta + \frac{ds}{da}\right) d\zeta. \tag{53}$$

If the distribution of material properties is symmetric relative to the central plane of the sheet, then Equation (53) becomes

$$M = T_0 \int_{0}^{1/2} W_Z(Z) Z dZ - T_0 \int_{-1/2}^{0} W_Z(Z) Z dZ = 2T_0 \int_{0}^{1/2} W_Z(Z) Z dZ. \tag{54}$$

Here Equation (52) has been taken into account.

Assume that one needs to find the solution of (46) at $a = a_f$. Let the interval $0 \le a \le a_f$ be subdivided into an arbitrary number of small segments by the points $a_i$. The values of $s$ and $ds/da$ at $a = a_i$ are denoted as $s_i$ and $s_i'$, respectively. It is convenient to choose these points a constant distance $\Delta a$ apart. Let the value of $a_i$, $s_i$, and $s_i'$ be known. Then, $a_{i+1} = a_i + \Delta a$. The value of $s_{i+1}$ can be approximated as

$$s_{i+1} = s_i + \frac{(s_i' + s_{i+1}')}{2} \Delta a. \tag{55}$$

Equation (46) at $a = a_{i+1}$ becomes

$$\int_{-s_{i+1}'}^{0} \frac{W(\mu)}{(\mu a_{i+1} + s_{i+1})} d\mu = \int_{-1}^{-s_{i+1}'} \frac{W(\mu)}{(\mu a_{i+1} + s_{i+1})} d\mu. \tag{56}$$

One can eliminate $s_{i+1}$ in (56) using (55). The resulting equation contains one unknown, $s_{i+1}'$. This equation should be solved numerically. Then, $s_{i+1}$ is readily determined from (55). Having found $s_{i+1}$ and $s_{i+1}'$, one can apply the procedure above to find the solution at $a_{i+1}$. This procedure should be repeated until $a_{i+1}$ becomes equal to $a_f$. It remains to find the input data for the first step. It is evident that $a_0 = 0$ and $a_1 = \Delta a$. It follows from (3) that $s_0 = 1/4$. The solution of (49) supplies $s_0'$. However, if the distribution of material properties is symmetric relative to the central plane of the sheet, then $s_0' = 1/2$, as follows from (52).

## 6. Unloading

Consider purely elastic unloading. Variations of the shape are neglected during this stage of the process. At the end of loading, $a = a_l$, $s = s_l$, and $m = m_l$. The corresponding values of the inside and outside radii are denoted as $r_{CD} = R_0$ and $r_{AB} = R_1$, respectively. The distribution of $\sigma_\zeta$ and $\sigma_\eta$ at $a = a_l$ is denoted as $\sigma_\zeta^{(l)}$ and $\sigma_\eta^{(l)}$. All the quantities introduced above can be calculated using the solution given in the previous sections.

The general solution for the increment of the principal stresses from the configuration corresponding to $a = a_l$ is independent of the solution at loading. Therefore, one may

adopt the solution for the plane strain bending under tension provided in [36] assuming the tensile force vanishes. In our nomenclature, this solution reads

$$
\frac{\Delta\sigma_\zeta}{2G} = V_0 \ln\left(\frac{r}{R_0}\right) - U_0\left[1 - \left(\frac{R_0}{r}\right)^2\right], \quad \frac{\Delta\sigma_\eta}{2G} = V_0\left[1 + \ln\left(\frac{r}{R_0}\right)\right] + U_0\left[1 + \left(\frac{R_0}{r}\right)^2\right],
$$
$$
U_0 = -\frac{\rho_0^2 \ln\rho_0\, m_l k}{2\left[\left(\rho_0^2-1\right)^2 - 4\rho_0^2 \ln^2\rho_0\right]}\frac{H^2}{R_0^2}, \quad V_0 = -\frac{\rho_0^2\left(1-\rho_0^2\right)m_l k}{2\left[\left(\rho_0^2-1\right)^2 - 4\rho_0^2 \ln^2\rho_0\right]}\frac{H^2}{R_0^2}
$$
(57)

where $\rho_0 = R_0/R_1$. Equation (4), in which $a$ should be replaced with $a_l$ and $s$ with $s_l$, supplies the dependence of $r$ on $\zeta$. The distribution of residual stresses is given by

$$
\sigma_\zeta^{res} = \sigma_\zeta^{(l)} + \Delta\sigma_\zeta \quad \text{and} \quad \sigma_\eta^{res} = \sigma_\eta^{(l)} + \Delta\sigma_\eta.
$$
(58)

After calculating the residual stresses, it is necessary to verify that the yield criterion is not violated in the range $-1 \leq \zeta \leq 0$. The corresponding condition follows from (7) in the form

$$
\left|\sigma_\eta^{res} - \sigma_\zeta^{res}\right| \leq T_0 W(\zeta).
$$
(59)

## 7. Practical Example

The through-thickness distribution of the coefficients involved in Hill's quadratic yield criterion [11] has been experimentally determined in rolled sheets of Al-Mg-Si alloy in [32]. The distribution is symmetric relative the central plane of the sheets. In our nomenclature, Table 1 represents the results from [32]. It is seen from this table that the function $W$ is non-monotonic in each half of the sheet. This discrete function is approximated to the following continuous function:

$$
W(\zeta) = 0.644 - 0.837\zeta - 4.888\zeta^2 - 8.103\zeta^3 - 4.051\zeta^4.
$$
(60)

It is worthy to note that the experimental data were first approximated by an even function of $\zeta + 1/2$ and then Equation (60) was derived. The numerical solution has been found using the general solutions described in Sections 3–6 and (60). In all calculations, $k = 0.001$.

**Table 1.** Through-thickness distribution of $W$.

| Surface ($\zeta$ = 0 and $\zeta$ = −1) | $\zeta$ = −1/4 and $\zeta$ = −3/4 | Center ($\zeta$ = −1/2) |
| --- | --- | --- |
| 0.644 | 0.659 | 0.6 |

It has been found that $a_e \approx 6 \cdot 10^{-5}$. Two plastic regions initiate in the vicinity of the inside and outside surface almost simultaneously and quickly propagate to the corresponding stress-free surface. This stage of the process is very short and is not significant for bending at large strains. The solution with three regions (two plastic regions and an elastic region between them) starts at $a = a_p \approx 2.7 \cdot 10^{-4}$. At $a \geq a_p$, the plastic region adjacent to the surface $\zeta = 0$ (plastic region 1) occupies the domain $\zeta_1 \leq \zeta \leq 0$ and the plastic region adjacent to the surface $\zeta = -1$ (plastic region 2) occupies the domain $-1 \leq \zeta \leq \zeta_2$. The elastic region occupies the domain $\zeta_2 \leq \zeta \leq \zeta_1$. Thus $\zeta = \zeta_1$ and $\zeta = \zeta_2$ are the elastic/plastic boundaries. Both $\zeta_1$ and $\zeta_2$ depend on $a$.

The distribution of $\sigma_\zeta$ and $\sigma_\eta$ in plastic region 1 follows from (22) and (24), and in plastic region 2 from (27) and (29). Using the boundary conditions in (14), one finds that $\sigma_\zeta^{(1)} = 0$, $\zeta^{(1)} = 0$, $\sigma_\zeta^{(2)} = 0$, and $\zeta^{(2)} = -1$. The stresses $\sigma_\zeta$ and $\sigma_\eta$ must be continuous across the elastic/plastic boundaries. This requirement is equivalent to the requirement

that $\sigma_\zeta$ and $\sigma_\zeta - \sigma_\eta$ are continuous across the elastic/plastic boundaries. Using (19), (20), (22), (24), (25), (27) and (29), one can represent the latter as

$$\ln^2[4(\zeta_1 a + s)] + 2kC = ak\int_0^{\zeta_1} \frac{W(\mu)}{(\mu a + s)}d\mu,$$

$$\ln^2[4(\zeta_2 a + s)] + 2kC = ak\int_{\zeta_2}^{-1} \frac{W(\mu)}{(\mu a + s)}d\mu, \tag{61}$$

$$2\ln[4(\zeta_1 a + s)] = kW(\zeta_1), \quad 2\ln[4(\zeta_2 a + s)] = -kW(\zeta_2).$$

One can eliminate $C$ between the first two equations. In the resulting equation, $\ln[4(\zeta_1 a + s)]$ and $\ln[4(\zeta_2 a + s)]$ can be eliminated using the third and fourth equations. Then,

$$\int_0^{\zeta_1} \frac{W(\mu)}{(\mu a + s)}d\mu - \int_{\zeta_2}^{-1} \frac{W(\mu)}{(\mu a + s)}d\mu = \frac{k}{4a}\left[W^2(\zeta_1) - W^2(\zeta_2)\right]. \tag{62}$$

Moreover, the third and fourth equations in (61) can be solved for $a$ and $s$. As a result,

$$a = \frac{1}{4(\zeta_1 - \zeta_2)}\left\{\exp\left[\frac{k}{2}W(\zeta_1)\right] - \exp\left[-\frac{k}{2}W(\zeta_2)\right]\right\},$$

$$s = \frac{1}{4(\zeta_2 - \zeta_1)}\left\{\zeta_2\exp\left[\frac{k}{2}W(\zeta_1)\right] - \zeta_1\exp\left[-\frac{k}{2}W(\zeta_2)\right]\right\}. \tag{63}$$

One can now eliminate $a$ and $s$ in (62) using (63). The resulting equation contains the two unknowns, $\zeta_1$ and $\zeta_2$. A numerical solution of this equation supplies the dependence of $\zeta_1$ on $\zeta_2$. Having found this dependence, $a$ and $s$ are readily calculated from (63) giving $\zeta_1$, $\zeta_2$, and $s$ as functions of $a$ in implicit form. Then, $C$ is determined from the first equation in (61). It has been found that $\zeta_1 \approx -0.189$ and $\zeta_2 \approx -0.811$ at $a = a_p$. It is more informative to use $H/r_{CD}$ as a time-like variable instead of $a$. Equation (5) allows one to replace $a$ with $H/r_{CD}$ with no difficulty. The variation of $\zeta_1$ and $\zeta_2$ with $H/r_{CD}$ is depicted in Figure 3. It is seen from this figure that the thickness of the elastic region decreases with $H/r_{CD}$ very quickly. In particular, this thickness is less than 0.8% of the sheet's initial thickness when $H/r_{CD} = 0.04$ (e.g., the inside radius of the sheet is 25 times larger than the sheet's initial thickness). The through-thickness distribution of $\sigma_\zeta$ and $\sigma_\eta$ at $H/r_{CD} = 0.04$ is shown in Figures 4 and 5, respectively. In these figures, X is the dimensionless distance from the inside surface of the sheet defined as

$$X = \frac{r - r_{CD}}{H}. \tag{64}$$

Using the numerical procedure described in Section 5, $s$ and $ds/da$ have been found as functions of $a$. Then, Equation (5) has been used to replace $a$ with $H/r_{CD}$. The dependence of $\zeta_n = -ds/da$ on $H/r_{CD}$ in the range $0 < H/r_{CD} \le 0.04$ is depicted in Figure 3. It is seen from this figure that the neutral line found from the rigid/plastic solution is located between the two elastic/plastic boundaries found from the elastic/plastic solution.

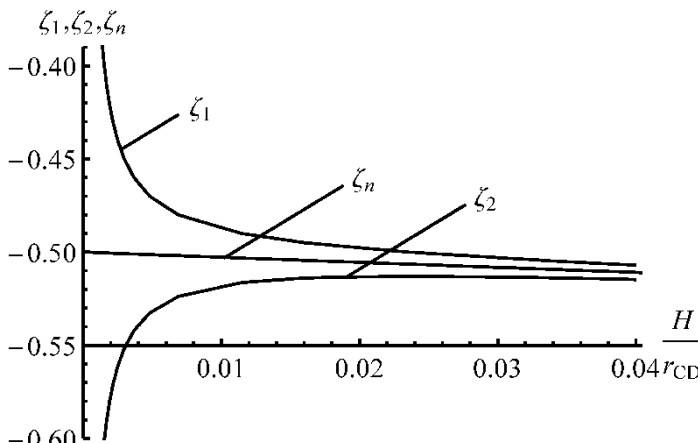

**Figure 3.** Variation of the locations of the elastic/plastic boundaries and the neutral line with $H/r_{CD}$ at the beginning of the process.

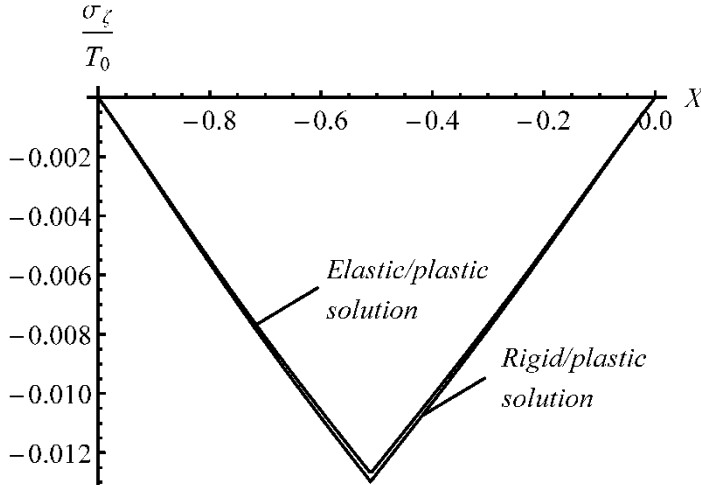

**Figure 4.** Distribution of the stress $\sigma_\zeta$ at $H/r_{CD} = 0.04$ found using the elastic/plastic and rigid/plastic solutions.

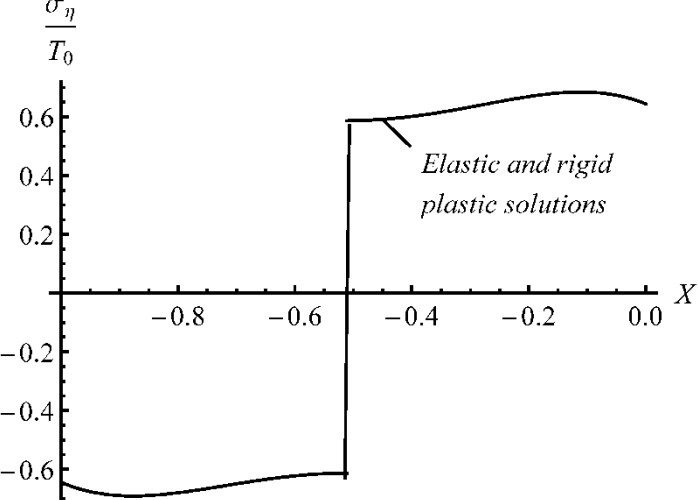

**Figure 5.** Distribution of the stress $\sigma_\eta$ at $H/r_{CD} = 0.04$ found using the elastic/plastic and rigid/plastic solutions.

The dependence of $\zeta_n = -ds/da$ on $H/r_{CD}$ in the range $0 < H/r_{CD} \leq 40$ is shown in Figure 6. It is seen from this figure that the location of the neutral line changes very quickly at the beginning of the process but gradually at large $H/r_{CD}$. The through-thickness distribution of $\sigma_\zeta$ and $\sigma_\eta$ at $H/r_{CD} = 0.04$ is shown in Figures 4 and 5, respectively. It is seen from Figure 4 that the difference between the elastic/plastic and rigid/plastic solutions is insignificant. The difference is most considerable near the neutral line in the rigid/plastic solution, and the rigid/plastic solution predicts a slightly higher value of $|\sigma_\zeta|$ than the elastic/plastic one. The difference between the elastic/plastic and rigid/plastic solutions is invisible in Figure 5. The through-thickness distribution of $\sigma_\zeta$ and $\sigma_\eta$ at several stages of the process found by means of the rigid/plastic solution is shown in Figures 7 and 8, respectively. The maximum value of $|\sigma_\zeta|$ significantly increases as the deformation proceeds.

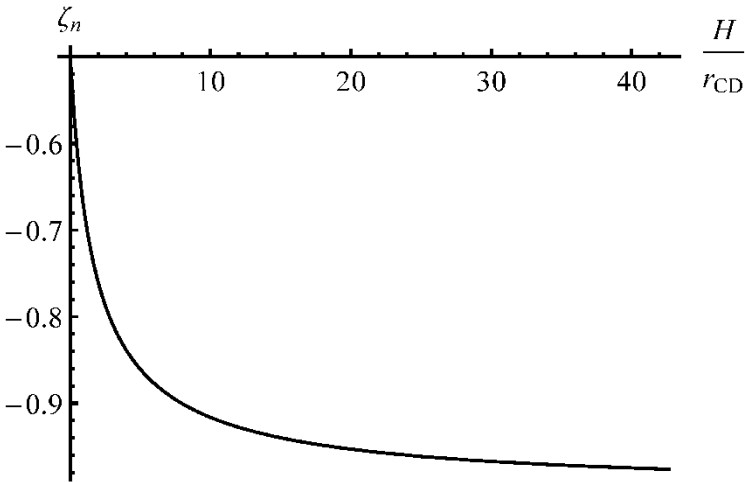

**Figure 6.** Variation of the location of the neutral line with $H/r_{CD}$.

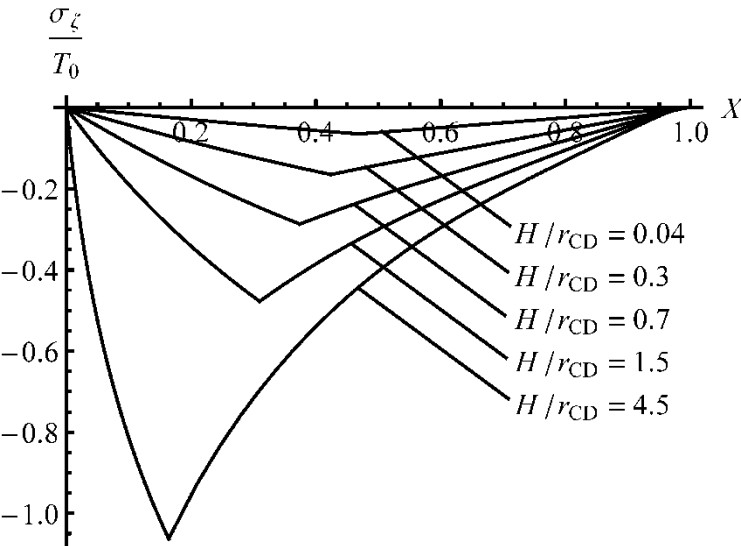

**Figure 7.** Distribution of the stress $\sigma_\zeta$ at several stages of the process.

The distribution of $\sigma_\eta$ has a weak local maximum in the vicinity of the outside surface, which may affect bendability. Having found the stress solution, one can calculate the bending moment using (18). The variation of the dimensionless bending moment with $H/r_{CD}$ is depicted in Figure 9. It is seen from this figure that the dimensionless bending moment changes very quickly at the beginning of the process but gradually at large $H/r_{CD}$. Its value attains a local minimum around $H/r_{CD} = 13$.

The procedure described in Section 6 has been applied to calculate the distribution of residual stresses. The methodology proposed in [34] has been adopted. Figures 4 and 5 justify the validity of this methodology. These calculations have shown that inequality (59) is not satisfied with relatively large values of $H/r_{CD}$, requiring consideration of reversed yielding. In particular, even if $H/r_{CD} = 0.04$, inequality (59) is not satisfied at a narrow region near the neutral line.

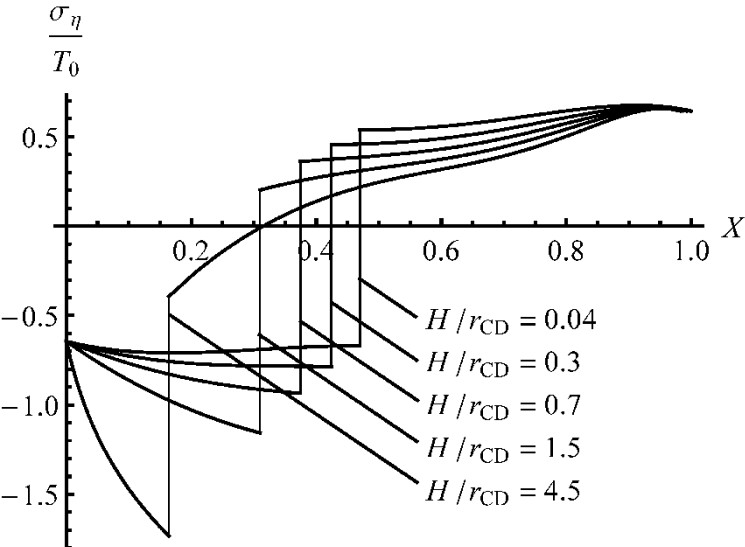

**Figure 8.** Distribution of the stress $\sigma_\eta$ at several stages of the process.

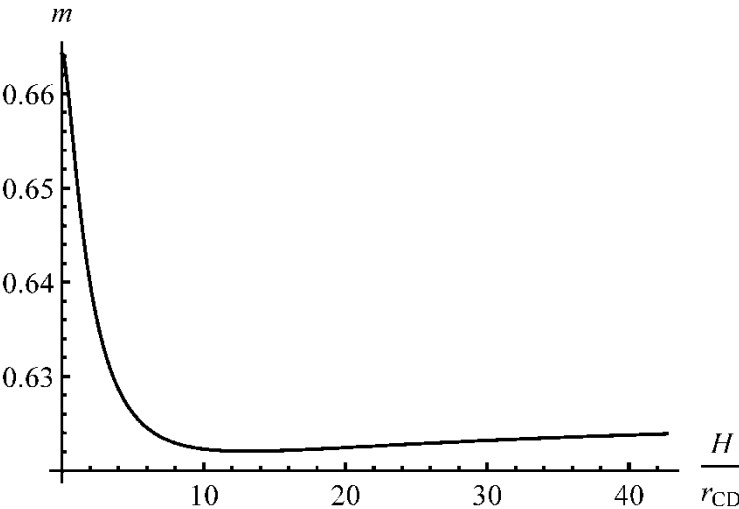

**Figure 9.** Variation of the dimensionless bending moment with $H/r_{CD}$.

## 8. Conclusions

A general semi-analytic solution for finite pure plane strain bending of a sheet made of incompressible material has been found. A distinguished feature of this solution is that the sheet is plastically anisotropic and inhomogenous. No restriction of the through-thickness distribution of plastic properties is imposed. Both elastic/plastic and rigid/plastic models have been considered. It has been shown that the difference between the elastic/plastic and rigid/plastic solutions is negligible, except at the very beginning of the process. Since the rigid/plastic solution is much simpler than the elastic/plastic one, it is recommended to use rigid/plastic models at large strains. It is possible even if residual stresses should be calculated. In this case, the methodology proposed in [34] can be adopted. The numerical

example uses real material properties provided in [32]. The through-thickness distribution of these properties is symmetric relative to the center line of the original sheet. The through-thickness stress distributions are illustrated in Figures 4, 5, 7 and 8. It is seen from Figure 5 and Equation (18) that the difference between the bending moment found using the elastic/plastic and rigid/plastic solutions is negligible if $H/r_{CD} \geq 0.04$, at least.

An advantage of the general solution is that it is valid for an arbitrary distribution of material properties. Therefore, in conjunction with experimental data, this solution can be readily used for identifying these properties.

It is crucial to predict springback in bending followed by unloading accurately [39]. The solution for purely elastic unloading is not valid for all cases considered, and it is necessary to consider the appearance of reversed yielding. The latter will be the subject of a subsequent investigation.

**Author Contributions:** Conceptualization, S.A.; writing, Y.-M.H.; formal analysis, E.L. All authors have read and agreed to the published version of the manuscript.

**Funding:** This research was made possible by the grants RFBR-19-51-52003 (Russia), MOST 108-2923-E-110-002-MY3 (Taiwan), and AAAA-A20-120011690136-2 (Russia).

**Institutional Review Board Statement:** Not applicable.

**Informed Consent Statement:** Not applicable.

**Conflicts of Interest:** The authors declare no conflict of interest.

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
