# Peer review of "Finite Pure Plane Strain Bending of Inhomogeneous Anisotropic Sheets"

_symmetry, doi:10.3390/sym13010145_

Round 1

Reviewer 1 Report

In the introduction of the article, the authors presented a study of literature that is connected to the main subject of the paper.
Authors highlight the research results in the available literature but in my opinion, the study of literature should be a little bit wider. In my opinion, the authors should highlight the results of research that are focused on the numeric method (For example https://doi.org/10.3390/en14020317) or analytical method in another industrial process (for example http://dx.doi.org/10.3390/ma13153317 ).

During the pre-review process was noted below pointed significant disadvantages:

1. In references, 42% of the papers were published more than 10 years ago. In my opinion, 50% of the papers should be published in a last 10 years.

2. Authors should highlight their own work.

3. Results of the analytical model are not compared with the results of empirical investigation. At the moment, it has not been proven to what extent the calculation results are consistent with the real values.

4. In poems 69, 188, 270, 274, 285, 288, 292, 293 & 298 add space symbol between "Figs." or "Fig." and the number of it.

5. In poem 304, replace the word Figure with the abbreviation Fig. as for the rest of the paper.

Author Response

In the introduction of the article, the authors presented a study of literature that is connected to the main subject of the paper.
Authors highlight the research results in the available literature but in my opinion, the study of literature should be a little bit wider. In my opinion, the authors should highlight the results of research that are focused on the numeric method (For example https://doi.org/10.3390/en14020317) or analytical method in another industrial process (for example http://dx.doi.org/10.3390/ma13153317 ).

During the pre-review process was noted below pointed significant disadvantages:

  1. In references, 42% of the papers were published more than 10 years ago. In my opinion, 50% of the papers should be published in a last 10 years.

We have added five recently published papers ([10], [14] – [16], [39]) in the revised manuscript. Unfortunately, the topic of the papers recommended by the reviewer, even remotely, is not related to the present paper's topic.  

  1. Authors should highlight their own work.

Our previous papers that are most closely related to the present paper are [27 -29], [31], and [36] (in the revised manuscript).

  1. Results of the analytical model are not compared with the results of empirical investigation. At the moment, it has not been proven to what extent the calculation results are consistent with the real values.

Direct comparison is impossible because of the lack of data. One has to bend a sheet made of the material tested in [32]. Otherwise, the data in Table 1 are not appropriate. In the case of closed-form solutions, any possible disagreement between the theoretical solution and experiment is due to the constitutive equations. In contrast to the example in Section 7, our general solution is valid for a large class of constitutive equations. It is an advantage of our solution. For this reason, the solution can be used for identifying the constitutive equations. We have mentioned it in Section 8 of the revised paper.

  1. In poems 69, 188, 270, 274, 285, 288, 292, 293 & 298 add space symbol between "Figs." or "Fig." and the number of it.

We have corrected the manuscript according to the instruction for authors (All Figures, Schemes and Tables should be inserted into the main text close to their first citation and must be numbered following their number of appearance (Figure 1, Scheme I, Figure 2, Scheme II, Table 1, etc.)).

  1. In poem 304, replace the word Figure with the abbreviation Fig. as for the rest of the paper.

We have corrected the manuscript according to the instruction for authors (All Figures, Schemes and Tables should be inserted into the main text close to their first citation and must be numbered following their number of appearance (Figure 1, Scheme I, Figure 2, Scheme II, Table 1, etc.)).

Reviewer 2 Report

The paper is written in good form, but some small doubts from references are open.

ref. 25. Elishakoff, I.; Pentaras, D.; Gentilini, C. Mechanics of functionally graded material structures, World Scientific: 385 Singapore, 2016; 340 p. DOI:10.1142/9505 Publisher: Publisher Location, Country, 2008; pp. 154–196.

Does the last information mean another reference? Some missing reference from paper?

Also, some references would be useful and bring attention to new possible readers.

For example:

Hung, L.T.Email Author, Dinh, V.T., Phuong, D.T., Kien, L.T. Effect of springback in DP980 advanced high strength steel on product precision in bending process. Acta Metallurgica Slovaca, Volume 25, Issue 3, 2019, Pages 150-157

Trzepiecinski, T. Effect of the plastic strain and drawing quality on the frictional resistance of steel sheets. Acta Metallurgica Slovaca, Volume 26, Issue 2, 2020, Pages 42-44

Also, papers from journal Symmetry?

Author Response

  1. 25. Elishakoff, I.; Pentaras, D.; Gentilini, C. Mechanics of functionally graded material structures, World Scientific: 385 Singapore, 2016; 340 p. DOI:10.1142/9505 Publisher: Publisher Location, Country, 2008; pp. 154–196.

Does the last information mean another reference? Some missing reference from paper?

It was a typo. We have corrected it in the revised manuscript.

  1. Also, some references would be useful and bring attention to new possible readers. 

Hung, L.T.Email Author, Dinh, V.T., Phuong, D.T., Kien, L.T. Effect of springback in DP980 advanced high strength steel on product precision in bending process. Acta Metallurgica Slovaca, Volume 25, Issue 3, 2019, Pages 150-157

Trzepiecinski, T. Effect of the plastic strain and drawing quality on the frictional resistance of steel sheets. Acta Metallurgica Slovaca, Volume 26, Issue 2, 2020, Pages 42-44

We have included these papers in the revised manuscript.

  1. Also, papers from journal Symmetry?

Papers [14 - 16] in the revised manuscript are from the journal Symmetry.

Reviewer 3 Report

In their manuscript, the authors demonstrate a semi-analytic solution for finite pure plane strain bending of a sheet made of incompressible material. The paper describes an analytical advance of rigid/plastic solution in the practical use at large strains. It is found that the difference between the elastic/plastic and rigid/plastic solutions is negligible, except at the very beginning of the bending process. Moreover,  the presented method has the high potential in the analysis of residual stresses.

Overall, this is a very good paper. The analytical advance to me is quite nice. In general, I find the manuscript very clearly written. For these reasons, I am strongly inclined to recommend it for publication in Symmetry.

Minor comments:

line 73: I think that AD and CB are straight, not "AC and BD are straight".

Eq. (1): "s" needs explanation. Each variable needs to be explained at first use.

Author Response

  1. line 73: I think that AD and CB are straight, not "AC and BD are straight".

 It was a typo there. We have corrected it in the revised manuscript.

  1. (1): "s" needs explanation. Each variable needs to be explained at first use.

We have moved the corresponding sentence to under Eq. (1).

Reviewer 4 Report

An interesting paper. I think it deserve to be published.

Author Response

Thank you for your comment.